# Explicit Granularity and Implicit Scale Correspondence Learning for Point-Supervised Video Moment Localization

## ABSTRACT

Video moment localization (VML) aims to identify the temporal boundary of the target moment semantically matching the given query. Existing approaches fall into three paradigms: fully-supervised, weakly-supervised, and point-supervised. Compared to other two paradigms, point-supervised VML strikes a balance between localization accuracy and annotation cost. However, it is still in its infancy due to the following two challenges: explicit granularity alignment and implicit scale perception, especially when facing complex cross-modal correspondences. To this end, we propose a Semantic Granularity and Scale Correspondence Integration (SG-SCI) framework aimed at modeling the semantic alignment between video and text, leveraging limited single-frame annotation information for correspondence learning. It explicitly models semantic relations of different feature granularities and adaptively mines the implicit semantic scale, thereby enhancing and utilizing modal feature representations of varying granularities and scales. SG-SCI employs a granularity correspondence alignment module to align semantic information by leveraging latent prior knowledge. Then we develop a scale correspondence learning strategy to identify and address semantic scale differences. Extensive comparison experiments, ablation studies, and necessary hyperparameter analyses on benchmark datasets have demonstrated the promising performance of our model over several state-of-the-art competitors.

## CCS CONCEPTS

• **Information systems** → **Novelty in information retrieval**; **Multimedia and multimodal retrieval**.

## KEYWORDS

Cross-modal Moment Localization; Cross-Modal Retrieval; Correspondence Learning

**ACM Reference Format:**
Anonymous Authors. 2024. Explicit Granularity and Implicit Scale Correspondence Learning for Point-Supervised Video Moment Localization. In *Proceedings of the 32nd ACM International Conference on Multimedia (MM'24), October 28-November 1, 2024, Melbourne, Australia.* ACM, New York, NY, USA, 10 pages. https://doi.org/10.1145/nnnnnnn.nnnnnnn

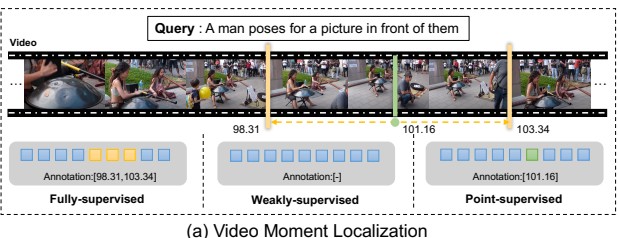

(a) Video Moment Localization

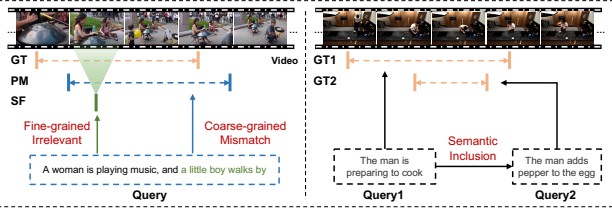

(b) Explicit Granularity Alignment and Implicit Scale Perception

**Figure 1: (a) Illustrative examples of Video Moment Localization (VML). (b) Illustration of explicit granularity alignment and implicit scale perception among VML. GT: Ground Truth; PM: Prediction Moment; SF: Supervised Frame. (Left) Fine-grained word-frame irrelevance in green and coarse-grained sentence-moment mismatch in blue. (Right) Illustration of the semantic inclusion relationships between textual and visual correspondences.**

## 1 INTRODUCTION

Video moment localization (VML), which refers to localizing a visual moment corresponding to a given textual query, is a fundamental task in video understanding. It benefits many important application scenarios, such as multimedia retrieval [45, 48] and smart human–computer interaction [37, 43].

Current efforts [14, 44, 50, 53] focus on addressing both fully and weakly supervised VML paradigms. As shown in Figure 1(a), the fully-supervised method requires the ground-truth moment annotations for training, which is laborious and time-consuming to obtain [29]. The weakly-supervised method is more flexible because it does not require moment annotations, but results in a significant drop in performance [10]. To balance accuracy with annotation cost, the point-supervised paradigm is proposed [21, 25]. Point-supervised methods require a single frame annotation from the ground truth, which is more practical and flexible compared to fully supervised annotation, costing only 1/6 as much [24].

Prior research [2, 4, 18, 35], to enhance video-language comprehension, can be classified into two primary approaches: granularity modeling and scale modeling. Granularity modeling involves delineating relationships among various features (such as frame-word and moment-sentence) through the **explicit** boundaries present in video-language, facilitating the differentiation of data across varying entity granularities. In contrast, scale modeling derives from the intricate semantic logics and inherent hierarchical structures within textual and visual domains. This approach acknowledges

**implicit** semantic scale in video and language interaction that exists within an alternative modal space.

To be specific, as shown in Figure 1(b), existing point-supervised labeling inevitably introduces two accompanied challenges: 1) **Explicit Granularity Alignment**. According to [10, 12], on the one hand, single-frame annotations are randomized within intervals, causing clear mismatches at a fine-grained level (frame-word); on the other hand, the incomplete information on interval annotations hinders the direct use of correspondences at a coarse-grained level (moment-query). Undoubtedly, this challenge poses a significant obstacle to effective language and visual alignment. 2) **Implicit Scale Perception**. Prior studies [9, 16] note that query sentences for the same video may vary in semantic scale. This issue is especially noticeable in the point-supervised approach. These methods struggle to capture temporal relationships within a video because of using only single-frame annotations, leading to a lack of effective contextual modeling constraints. As a result, the essential semantic scale information remains unmodeled, impeding temporal learning and video comprehension.

To navigate these challenges, we introduce a novel **S**emantic **G**ranularity and **S**cale **C**orrespondence **I**ntegration (**SG-SCI**) framework that integrates a *Granularity Correspondence Alignment (GCA)* module and a *Scale Correspondence Learning (SCL)* strategy. Firstly, the GCA module is engineered to enhance the interaction of video-language. By employing a fine-grained alignment approach, it establishes a more detailed and comprehensive mapping between video frames and textual descriptions. This module not only facilitates a deeper understanding of the video content but also ensures that even the less prominent frames find relevance in the corresponding textual narrative. Secondly, the SCL strategy addresses the disparity in temporal scale. It is designed to learn latent semantics adaptively across varying temporal scales, enabling the model to assimilate and correlate extensive video sequences with succinct textual queries. This strategy ensures a more robust and contextually aware matching process, thereby enhancing the accuracy and adaptability. Consequently, this framework offers a more nuanced and adaptable solution for point-supervised VML.

To the best of our knowledge, it is the **first work** on integrating explicit granularity alignment and implicit scale perception into point-supervised VML. Our approach effectively mitigates the granularity and scale-related challenges by the synergy of the innovative component and strategy. It boosts the precision of aligning video moments to text queries and enhances the model's robustness.

In summary, the main contributions are as follows:

- **Model Contribution.** We introduce an innovative Granular Correspondence Alignment module. Specifically, it is designed to improve the explicit correspondence relation across varying granularities among different modalities.
- **Strategy Contribution.** Under the framework of point-supervised, we develop a Scale Correspondence Learning strategy, which is pivotal in capturing the implicit semantic scale in correspondence learning.
- **Experimental Contribution.** Extensive experiments on two benchmarks, *i.e.*, Charades-STA [12] and TACoS [33], validate the effectiveness and superiority of our model. The codes and settings are released at https://anonymous.4open.science/r/SG-SCI.

## 2 RELATED WORK

### 2.1 Video-Language Modeling

Current research can be divided into two main categories depending on the use of a modeling approach: feature granularity modeling and semantic scale modeling. In terms of entities, the former is explicit while the latter is implicit.

Early explicit modeling approaches [1, 12] mainly used global sentence-level alignment to enhance the semantics of visual features, but this approach overlooked fine-grained semantics such as words. As a result, several studies [4, 35, 49] have investigated various word-frame interactions using attention mechanisms to capture the relationships between visual cues and textual queries. Recent research [38, 47, 52] has recognized the significance of local phrase patterns or tokens in sentences for video moment retrieval.

Compared to explicit modeling, implicit scale modeling [2, 11, 18, 19, 39] utilizes multi-scale relations to achieve better grounding results. These relations mainly exist at the video-level and language-level. However, all grounding methods overlook the possibility of different semantic scales within the same level (e.g., sentence level). Accordingly, DualMIL [9] extends multiple instance learning into a two-level framework.

Although the aforementioned modeling strategies are effective, most methods do not fully utilize their potential when dealing with limited supervisory information. Therefore, our focus is on extracting both implicit and explicit information to accurately locate boundaries in point-supervised VML.

### 2.2 Video-Language Correspondence Learning

In the domain of video-language learning, the misalignment between textual descriptions and corresponding visual content introduces significant challenges, necessitating the adoption of correspondence learning as an innovative approach [15, 17, 20]. This paradigm shift is epitomized by the introduction of MIL-NCE [31], which pioneers the strategy of aligning video clips with adjacent sentences to diminish the effects of misalignment. In contrast, Tang *et al.* [40] improves video description quality by integrating an external image captioning model, emphasizing data-driven enhancement over immediate error correction.

Our method differs itself from preceding endeavors through two principal innovations. Firstly, we go beyond the traditional focus on either model architectures or supervisory strategies by enhancing improvements through the synergistic interaction between model and strategy learning. Secondly, we expand the use of correspondence learning beyond segmented multi-modal data. This pioneering adaptation allows for precise moment localization with point-supervised for the first time.

### 2.3 Video Moment Localization

Different from existing fully and weakly-supervised VML studies [12, 14, 27, 30, 44, 50, 53], point-supervised VML strikes a delicate balance between annotation efforts and model performance by leveraging a single frame from the localization moment [10, 21, 25]. Compared to fully-supervised VML, point-supervised significantly reduces the cost of annotation data and it provides more comprehensive information than weakly supervised learning. By eliminating

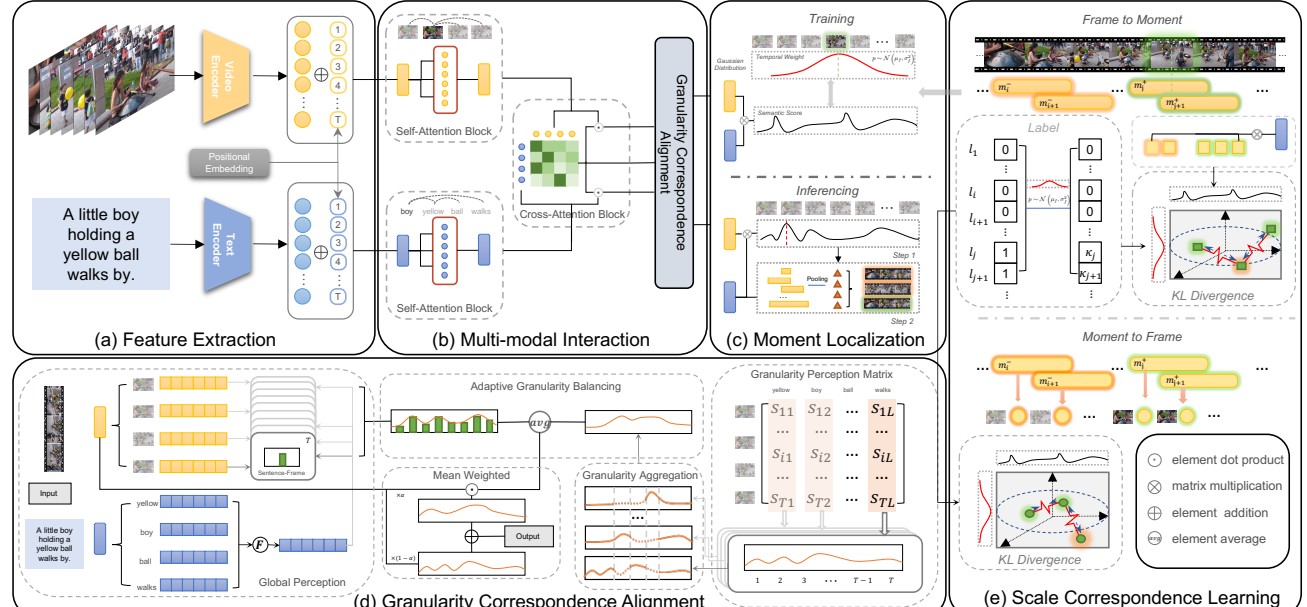

Figure 2: Schematic illustration of the proposed SG-SCI model. We first perform (a) Feature Extraction on both video and language separately, followed by (b) Multi-modal Interaction, which combines the obtained cross-attention features and similarity information, feeding them into the (d) Granularity Correspondence Alignment module. After the interaction, the features are input into the (c) Moment Localization module. During the training phase, we apply a (e) Scale Correspondence Learning strategy to compare semantics at diverse scales.

the need for precise start and end timestamps for the target moment, a quick "glance" at the video and the selection of a single frame are often sufficient, making point-supervised methods applicable to scenarios with incomplete annotation information [46]. The concept of point-supervised was initially introduced by Bearman *et al.* [3] in the context of semantic segmentation tasks and Cui *et al.* [10] applied the concept of point-supervised to VML and introduced the ViGA model, which aligns Gaussian distributions generated from supervised frame positions with cross-modal attention. Subsequent works have built upon ViGA's foundation, D3G [25] is proposed to align features of sentence-moment pairs and dynamically mitigate annotation bias using Gaussian distributions, while CFMR [21] develops a concept-driven multi-modal alignment mechanism to circumvent the need for cross-modal interaction modules during the inference process.

Despite efforts to enhance retrieval performance, existing work still falls short in effectively modeling the semantic granularity relationship between two modalities. Furthermore, due to the constraints of the supervised information, it struggles to capture the overall action changes within a moment.

## 3 METHODOLOGY

This section commences by defining the problem and outlining the core pipeline of point-supervised video moment localization. After this, we will explore the insights behind our innovative *Granularity Correspondence Alignment* method, highlighting its main elements. Finally, we conclude with an exposition of our *Scale Correspondence Learning* strategy, showing its significance in our research. Figure 2 presents an overview of our proposed SG-SCI.

### 3.1 Problem Definition

In this work, we aim to address the challenge of cross-modal video localization under point-supervised. The dataset comprises a series of quadruples and is denoted as $\mathcal{D}$, which is expressed as follows,

$$\mathcal{D} = \{(\mathcal{V}_i, Q_i, \tau_i^s, \tau_i^e) \mid i = 1 \text{ to } N\}, \tag{1}$$

where $\mathcal{V}_i$ represents an untrimmed video, $Q_i$ denotes the associated query sentence, and $\tau_i^s, \tau_i^e$ are the start and end timestamps respectively. The target video moment aligns with the query $Q_i$.

Contrasting with a fully-supervised paradigm, our training is based on a collection of triplet annotations, symbolized as $\mathcal{A} = \{(\mathcal{V}_i, Q_i, \tau_i^m) \mid i = 1 \text{ to } N\}$, where $\tau_i^m$ signifies a randomly selected point within the interval defined by $\tau_i^s$ and $\tau_i^e$. During the inference phase, the objective is to precisely localize the relevant video moment for each query within the dataset $\mathcal{D}$.

### 3.2 Pipeline of Point-supervised VML

To fulfil the task of Point-supervised VML, we first introduce the pipeline which can be categorized into three parts: *Feature Extraction*, *Multi-modal Interaction*, and *Moment Localization*.

*3.2.1 Feature Extraction.* Before multi-modal interaction, we extract the visual and textual features, and then incorporate learnable positional embedding into the extracted features.

**Visual Representation.** Given an untrimmed video $\mathcal{V}$, our approach utilizes a pre-trained CNN, i.e., C3D [41] or I3D [6], to extract visual features to ensure fair comparison. The extracted features are then processed through a fully-connected (FC) layer, obtaining a visual representation denoted as $V = \{v_1, v_2, \ldots, v_T\}$,

where $T$ represents the number of frames sampled at intervals in the video and $v_i$ denotes the $i$-th frame.

To perceive the time of the video, we incorporate learnable Positional Embedding (PE) into the model and get enhanced visual representation $\bar{V} \in \mathbb{R}^{T \times d}$, where $d$ represents the dimension of the visual representation.

**Text Representation.** For a given query $Q$, we employ the pre-trained Glove model [32] to extract textual features. A bi-directional gated recurrent unit (Bi-GRU) is employed to obtain the latent sequential order of original sentence, and get new textual representation $Q = \{q_1, q_2, \ldots, q_L\}$, where $L$ indicating the number of words in the query. Similar to the visual representation, PE is also added to the text representation, obtaining the enhanced features $\bar{Q} \in \mathbb{R}^{L \times d}$.

*3.2.2 Multi-modal Interaction.* To capture the intra-model semantics, we utilize $\bar{V}$ and $\bar{Q}$ through the self-attention layer [42] to generate the intra-modal representation $\hat{V}$ and $\hat{Q}$. Similarly, we use the cross-modal multi-head attention to obtain the cross-modal representation $\tilde{V} = Attn(\hat{Q}, \hat{V})$ and $\tilde{Q} = Attn(\hat{V}, \hat{Q})$, where $Attn(\cdot)$ is formulated as follows,

$$
\begin{aligned}
Attn(\hat{Q}, \hat{V}) &= \text{softmax}(\frac{\hat{Q}\hat{V}^T}{\sqrt{d}})\hat{V}, \\
Attn(\hat{V}, \hat{Q}) &= \text{softmax}(\frac{\hat{V}\hat{Q}^T}{\sqrt{d}})\hat{Q}.
\end{aligned}
\tag{2}
$$

Subsequently, to serve the subsequent modules, we take out the last layer of cross-modal attention scores as the cross-modal attention scores $A^{q \to v}$ and $A^{v \to q}$, which will be further illustrated in 3.3. We input the features and attention scores into our proposed granularity correspondence alignment module to obtain the multi-granularity representation $\tilde{V}_f$ and $\tilde{Q}_w$:

$$
\begin{aligned}
\tilde{V}_f &= \text{GCA}(\tilde{V}, \tilde{Q}, A^{q \to v}), \\
\tilde{Q}_w &= \text{GCA}(\tilde{Q}, \tilde{V}, A^{v \to q}),
\end{aligned}
\tag{3}
$$

where GCA($\cdot$) is the granularity correspondence alignment module. So far, we have obtained the multi-granularity representation $\tilde{V}_f$ and $\tilde{Q}_w$ via multi-modal interaction. The representation is compound of multi-level granularities information, assisting cross-modal semantic alignment. In order to integrate the semantic information of the whole sentence, we use the max-pooling function $F(\cdot)$ to obtain the sentence representation $\tilde{Q}_s$ as follows,

$$
\tilde{Q}_s = \text{F}(\tilde{Q}_w).
\tag{4}
$$

*3.2.3 Moment Localization.* Moment localization aims to locate the start and end timestamps based on the obtained representation. In the training stage, unlike fully-supervised scenario, a random point within the target video moment is accessed while it is absent during inference. Because of the divergence between the two stages, we will elaborate on both of them subsequently.

**Training Stage.** In the training process, we use sliding windows to slice $\tilde{V}_f$, and generate candidate moments. Gaussian distribution is utilized to measure the moments' temporal features and cosine similarity of semantic features. The temporal information is defined

as follows,

$$
G_i = \frac{1}{\sqrt{2\pi}\sigma} \exp\left(-\frac{\left((i - \tau^m) \cdot \frac{2}{L_v - 1}\right)^2}{2\sigma^2}\right),
\tag{5}
$$

where $i$ is the $i$-th frame, $\tau^m$ is the index of the supervised frame, $\sigma$ is a hyperparameter, and $L_v$ is the length of the video. The semantic information $S_i$ is the cosine similarity between $\tilde{V}_f(i)$ and $\tilde{Q}_s$.

These temporal and semantic analyses are integrated and applied in our scale correspondence learning strategy, detailed in Sec. 3.4.

**Inference Stage.** The inference involves two primary steps, consisting of identifying the key point that best matches query and expand from this point to get the predicted video moment most similar to the query.

## 3.3 Granularity Correspondence Alignment

In this section, we introduce a novel granularity correspondence alignment module within the framework of point-supervised VML, which is designed to adaptively regulate the granularity relationship between different modalities. By exploring semantic relationships across modalities, the module enriches the original modal representation and supplements existing semantic alignment with information across various granularities, ensuring symmetry in intermodal interactions.

Unlike existing methods that unconsciously interact the smallest granularity units (frames and words) in representations across modalities, our main insights derive from exploiting the attention-perceived matrix and utilizing it to capture potential cross-modal semantic granularity information. Specifically, we use the matrix consisting of cross-modal attention scores as a granularity perception matrix as follows,

$$
A^{q \to v} = \begin{bmatrix}
s_{11} & s_{12} & \cdots & s_{1L} \\
\cdots & \cdots & \cdots & \cdots \\
s_{i1} & s_{i2} & \cdots & s_{iL} \\
\cdots & \cdots & \cdots & \cdots \\
s_{T1} & s_{T2} & \cdots & s_{TL}
\end{bmatrix},
\tag{6}
$$

where $A^{q \to v} \in \mathbb{R}^{T \times L}$ is the cross-modal attention score matrix, $T$ and $L$ are the number of frames and words, respectively, and $s_{ij}$ is the attention score between the $i$-th frame and the $j$-th word. The differences of scores in each column represent how much attention the word pays to the video frame. Therefore, we consider using max function to aggregate the information in each row to obtain the potential prior distribution $\tilde{V}_p \in \mathbb{R}^T$ of the complete query with respect to the video frames, formulated as,

$$
\tilde{V}_p = \max(A^{q \to v}) = [\max(s_{1\cdot}), \max(s_{2\cdot}), \ldots, \max(s_{T\cdot})]^T.
\tag{7}
$$

Next, we calculate the cosine similarity between the query and the original video frames to obtain the prior distribution of the complete query with respect to the video frames,

$$
\tilde{V}_g = cos(\tilde{V}, \tilde{Q}_s),
\tag{8}
$$

where $cos(\cdot)$ is the cosine similarity function. Subsequently, we use the finest granularity information from attentional perception and pooled global granularity information, which are averaged and used as adaptive visual granularity perceptual features $\tilde{V}_c = \frac{1}{2}(\tilde{V}_g + \tilde{V}_p)$, In this way, we can obtain the visual granularity perception vector

$\tilde{V}_c$, which can be used to enhance the original visual representation. On this basis, we use the obtained features to remodel the cross-modal fusion with text feature,

$$\tilde{V}_a = \tilde{V}_c \odot \tilde{V}_f. \tag{9}$$

After reaggregation, we integrate self-attention-based fine-grained features with multi-scale visual perception features. This enhances fine-grained entity alignment and focuses on overall perception, thereby improving representation. We then apply these features for cross-modal semantic interactions, using prior semantic knowledge to refine visual-word correlations.

Finally, we use the weighted mean approach in order to control the original modal feature information, thus preventing noise interference due to excessive introduction of cross-modal information as follows,

$$\tilde{V}_f = \alpha \tilde{V} + (1 - \alpha)\tilde{V}_a, \tag{10}$$

where $\alpha$ is the mean weighted factor. We use the same strategy applied to another modality. Therefore, we can obtain the final cross-modal representation $\tilde{V}_f$ and $\tilde{Q}_w$ via the multi-modal interaction.

## 3.4 Scale Correspondence Learning

To improve the model's understanding ability under different scales of moments, we utilize potential frame-moment correspondence semantic information, as the single-frame supervised information alone is not sufficient for effective moment semantics. To be specific, the target of optimization consists of three parts, the first part models the semantic and temporal information of the fused features from a global perspective. The second part uses point annotation to compare the differences between potential positive and negative samples in different intervals to effectively capture the information about the change of actions within the intervals. The third part aims to exploit the single point information in the global moment that is most similar to the query, which in turn enriches the priori knowledge and enhances the guidance of cross-modal semantics.

### 3.4.1 Global Alignment Loss.
We take the cross-entropy loss to force the representation information we get to be close to the information provided by the supervised frames in terms of temporal distance and semantic distance,

$$L_g = -\sum_{i=1}^{T} G_i \log S_i, \tag{11}$$

where $G_i$ is the Gaussian distribution weight of the $i$-th frame, and $S_i$ is the semantic similarity score of the $i$-th frame.

### 3.4.2 Frame-moment Correspondence Loss.
Due to lacking of boundary annotations, a single point's information inadequately represents the interval matching the query. This makes us to consider using video moments that encompass point annotation as potential positives. We construct binary labels to facilitate interval perception: moments containing the point annotation are labeled 1, while others are labeled 0, as follows,

$$\mathbf{y}_i = \begin{cases} 1, & \tau^m \in \mathbf{M}_i, \\ 0, & otherwise, \end{cases} \tag{12}$$

where $\mathbf{M}_i$ is the $i$-th moment by sliding window. Due to the latent differences in temporal information of these moments, we introduce

Gaussian distribution weights to further construct the soft labels $\tilde{\mathbf{y}}_i$, which can be formulated as:

$$\tilde{\mathbf{y}}_i = \mathbf{y}_i \mathcal{W}(\mathbf{M}_i), \tag{13}$$

where $\mathcal{W}(\cdot) = G(s) * G(e)$, $G(s)$ and $G(e)$ are the Gaussian distribution weights at the beginning and end of $\mathbf{M}_i$, respectively. Since it is inevitable that there will be multiple positive sample moments belonging to a query, it is inappropriate to view similarity score learning as a *1-in-N* classification problem with cross-entropy loss. To this end, we utilize the Kullback-Leibler divergence to construct frame-moment correspondence loss $L_c^{f \to m}$, as follows,

$$L_c^{f \to m} = \sum_{i=1}^{N} \tilde{\mathbf{y}}_i \log \frac{\tilde{\mathbf{y}}_i}{\mathbf{S}_i}, \tag{14}$$

where $\mathbf{S}_i$ is the cosine similarity calculated by moment $\mathbf{M}_i$ and query feature $\tilde{Q}_s$.

### 3.4.3 Moment-frame Correspondence Loss.
In the beginning of location process, we focus on identifying a point that align most closely with the query, enabling our model to discern and emphasize subtle points' differences. We separate essential information from positive sample moments, concentrating on feature variances. Specifically, we extract detailed point data reflecting semantic labels across moments, leveraging this to enrich point annotation with inherent disparities.

This process can be represented as follows,

$$\mathbf{K}_i = \tilde{V}_f(\theta), \theta = \arg \max_{j \in [s,e]} S_j, \tag{15}$$

where $s$ and $e$ are the start and end index of the moment $\mathbf{M}_i$.

For label construction, we use both hard sample labels and soft labels with Gaussian weights, just as the Eqn. (12) and (13). We then employ KL divergence to analyze differences between positive and negative samples, using these insights to refine our alignment process, as follows,

$$L_c^{m \to f} = \sum_{i=1}^{N} \tilde{\mathbf{y}}_i \log \frac{\tilde{\mathbf{y}}_i}{\mathbf{K}_i}. \tag{16}$$

By combining the global loss $L_g$ and correspondence loss $L_c$, we obtain the final loss for model optimization, as follows,

$$L = L_g + \beta L_c^{f \to m} + \gamma L_c^{m \to f}, \tag{17}$$

where $\beta$ and $\gamma$ are used to balance the focus between moments and frames, avoiding the model over underscore the sample information.

## 4 EXPERIMENTS

This section will begin by presenting the experimental settings. Subsequently, we will conduct comparative experiments, comprehensive ablation studies, and evaluate the effectiveness of SG-SCI to answer the following three research questions (RQs):

- **RQ1**: Is our proposed SG-SCI able to outperform several state-of-the-art competitors on VML?
- **RQ2**: Is each component of our SG-SCI helpful for boosting the localization performance?
- **RQ3**: How do hyperparameters affect model capability?

## 4.1 Datasets

The experiments that follow are conducted on two benchmark datasets: Charades-STA [12] and TACoS [33].

**Charades-STA** [12]: It is constructed on the Charades dataset [36] and contains 16,128 "moment-query" pairs with an average video duration of 30 seconds. Following the standard split strategy [12], we divided the dataset into 12,408 and 3,720 "moment-query" pairs for training and testing, respectively.

**TACoS** [33]: It is built upon MPII Cooking Compositive dataset [34] and only covers cooking activities that contain pairs of queries with very similar visual information. It consists of 127 untrimmed videos with the average duration of 320 seconds and 18,818 queries.

In particular, we use point-supervised information instead of boundary supervision information in the original dataset setting, just like existing effort [10], where the point-supervised information comes from a random point in the boundary.

## 4.2 Experimental Settings

*4.2.1 Evaluation Metrics.* Following the existing work [10, 21, 25], we choose $Rn@m$ and mean averaged IoU (mIoU) as protocols to evaluate the performance of moment localization. To be specific, $Rn@m$ refers to the percentage of predictions for which the temporal Intersection over Union (IoU) surpasses the thresholds $m$ within the top-$n$ of the sorted results, and mean averaged IoU (mIoU) represents the average IoU across all test samples. Since our method localizes the target moment with the highest coordinate probability, we set $n = 1$ and $m \in \{0.3, 0.5, 0.7\}$. The higher $Rn@m$ and mIoU are better.

*4.2.2 Implementation Details.* In our work, we employ the pre-trained I3D [6] and C3D [41] network to extract visual features from Charades-STA [12] and TACoS [33] respectively. Consistent with prior research [10], we keep the word embedding module fixed and utilize the 840B GloVe [32] to construct a comprehensive vocabulary. The model dimension $d_{model}$ is set to 512, and we employ AdamW [28] with a learning rate of $1e^{-4}$, which decays by half when reaching a plateau during training. By default, we set $\alpha = 0.3$, $\beta = \gamma = 0.1$, and $\sigma = 0.04$ in both datasets. The batch sizes for the Charades-STA and TACoS are empirically set to 512 and 64, respectively. All experiments are conducted on a NVIDIA GeForce RTX 4090 with 24GB memory.

## 4.3 Performance Comparison (RQ1)

In this subsection, we compare the proposed method with different kinds of state-of-the-art methods, including fully-supervised methods (**2D-TAN** [51], **SS** [11], **FVMR** [13], **ADPN** [7], **MS-DETR** [22]), weakly-supervised methods (**SCN** [26], **CNM** [54], **CWG** [8], **CPL** [55], **IRON** [5], **PPS** [23]) and point-supervised methods (**ViGA** [10], **PSVTG** [46], **CFMR** [21], **D3G** [25]). To ensure the validity of the comparison, the majority of the selected methods are drawn from the last three years. The comparisons on Charades-STA and TACoS are encapsulated in Table 1 and Table 2 respectively. Based on the data presented, the following observations can be derived:

- **Our method mines more correspondence information compared with existing point-supervised methods.** While traditional point-supervised methods focus on the interaction of

**Table 1: Performance comparison on Charades-STA with different supervision methods. Bold means the best result in point-supervised method and _underline_ means the second best.**

| Type | Method | R1@0.3 | R1@0.5 | R1@0.7 | mIoU |
|---|---|---|---|---|---|
| Fully-supervised | 2D-TAN [AAAI20] [51] | - | 50.62 | 28.71 | - |
| | SS [ICCV21] [11] | - | 56.97 | 32.74 | - |
| | FVMR [ICCV21] [13] | - | 55.01 | 33.74 | - |
| | ADPN [ACMMM23] [7] | 70.35 | 55.32 | 37.47 | 51.15 |
| Weakly-supervised | CWG [8] | 43.41 | 31.02 | 16.53 | - |
| | CPL [CVPR22] [55] | 66.40 | 49.24 | 22.39 | - |
| | IRON [CVPR23] [5] | 70.28 | 51.33 | 24.31 | - |
| | PPS [AAAI24] [23] | 69.06 | 51.49 | 26.16 | - |
| Point-supervised | ViGA [SIGIR22] [10] | **71.21** | 45.05 | 20.27 | _44.57_ |
| | PSVTG [TMM22] [46] | 60.40 | 39.22 | 20.17 | 39.77 |
| | CFMR [ACMMM23] [21] | - | _48.14_ | _22.58_ | - |
| | D3G [CVPR23] [25] | - | 43.82 | 20.46 | - |
| | SG-SCI (Ours) | _70.30_ | **52.07** | **27.23** | **46.77** |

**Table 2: Performance comparison on TACoS with different supervision methods. Bold means the best result in point-supervised method and _underline_ means the second best.**

| Type | Method | R1@0.3 | R1@0.5 | R1@0.7 | mIoU |
|---|---|---|---|---|---|
| Fully-supervised | 2D-TAN [AAAI20] [51] | 37.29 | 25.32 | - | - |
| | SS [ICCV21] [11] | 41.33 | 29.56 | - | - |
| | FVMR [ICCV21] [13] | 41.48 | 29.12 | - | - |
| | MS-DETR [ACL23] [22] | 47.66 | 37.36 | 25.81 | 35.09 |
| Weakly-supervised | SCN [AAAI20] [26] | 11.72 | 4.75 | - | - |
| | CNM [AAAI22] [54] | 7.20 | 2.20 | - | - |
| | CPL [CVPR22] [55] | 11.42 | 4.12 | - | - |
| Point-supervised | ViGA [SIGIR22] [10] | 19.62 | 8.85 | 3.22 | 15.47 |
| | PSVTG [TMM22] [46] | 23.64 | 10.00 | 3.35 | _17.39_ |
| | CFMR [ACMMM23] [21] | 25.44 | _12.82_ | - | - |
| | D3G [CVPR23] [25] | _27.27_ | 12.67 | _4.70_ | - |
| | SG-SCI (Ours) | **37.47** | **20.59** | **8.27** | **23.83** |

sentence-moments [25], our method involves multi-granularity interactions among complex video, concise textual descriptions, and point annotation. Therefore, the model can consider the interaction between sentences and point annotation at multiple granularities during the training process, resulting in better cross-modal information interaction. As illustrated in Table 1, in the Charades-STA dataset, SG-SCI achieves **4.65%** and **6.77%** higher than CFMR [21] and D3G [25] in $R1@0.7$, respectively.

- **SG-SCI reduces the performance disparity between fully-supervised methods and point-supervised methods.** Lacking of annotated timestamps, the positioning accuracy of point-supervised methods is lower compared to existing supervised learning methods. By learning features across different scales, SG-SCI exploits the information contained in the supervised frames, to some extent compensating for the performance gap caused by different supervised information. In the Charades-STA dataset, our method even outperforms 2D-TAN [51], fully-supervised method, by 1.45% in terms of $R1@0.5$.

- **Trivial annotation cost contributes to localization.** The existence of frame annotation provides the model with a clear direction during iterative convergence, and our proposed correspondence module enables the model to continuously converge during gradient backpropagation based on the multi-granularity information from query and frame annotation. As evidenced in Table 1 and Table 2, our approach consistently outperforms the cutting-edge weakly-supervised models.

(a) Component Ablation.  (b) GCA Ablation.  (c) Mean Weighted Factor.

Figure 3: Impact of (a) GCA and SCL, (b) multiple granularities in GCA, and (c) the mean weighted factor $\alpha$ in GCA. Evaluation performed on Charades-STA.

Table 3: Comparison of different loss ablation for our framework. ✔ means retaining it and ✘ means removing it.

| Global | F-M | M-F | R1@0.3 | R1@0.5 | R1@0.7 | mIoU |
|--------|-----|-----|--------|--------|--------|------|
| ✔ | ✘ | ✘ | 66.83 | 37.74 | 14.14 | 40.81 |
| ✘ | ✔ | ✘ | 50.56 | 27.96 | 10.27 | 31.04 |
| ✘ | ✘ | ✔ | 41.13 | 23.87 | 9.46 | 26.33 |
| ✘ | ✔ | ✔ | 47.50 | 25.59 | 9.70 | 29.13 |
| ✔ | ✘ | ✔ | 70.00 | 44.70 | 19.81 | 44.04 |
| ✔ | ✔ | ✘ | 69.89 | 42.07 | 17.50 | 43.28 |
| ✔ | ✔ | ✔ | 70.30 | 52.07 | 27.23 | 46.77 |

- **SG-SCI can handle more challenging task.** In contrast to Charades-STA, TACoS contains longer videos with shorter retrieved moments, and each frame within the video exhibits a high degree of semantic approximation, demanding a heightened ability from the model to perceive boundaries and differentiate similar moments. Consequently, existing weakly-supervised methods may struggle to address this scenario, resulting in a significant absence of the $R1@0.7$ metric in Table 2. However, compared to existing point-supervised methods, like ViGA [10] and D3G [25], our approach exhibits great performance in $R1@0.7$.

## 4.4 In-depth Analysis (RQ2 & RQ3)

To demonstrate each component of SG-SCI, we conducted extensive ablation experiments on Charades-STA.

### 4.4.1 Ablation Study.

- **Impact of GCA and SCL.** To discuss whether the proposed GCA and SCL can achieve better performance, we simplify the pipeline of SG-SCI and set it as the baseline. Specifically, after multi-modal interaction, we train the model by calculating the KL divergence of the similarity distribution between the query and each frame with a Gaussian distribution. To demonstrate the effectiveness of SCL, we conduct experiments by setting the soft labels to 1. From the Figure 3(a), it can be seen that both GCA and SCL effectively enhance the localization ability of the baseline, especially in $R1@0.7$.

- **Effect of multiple granularities in GCA.** As described in Sec 3.3, GCA not only helps to understand the video contents more deeply but also ensures that less significant frames have

relevance in the corresponding textual narrative. We validate the effectiveness of multi-granularity information interaction on cross-modal representation, with specific results shown in Figure 3(b). It can be seen that the multi-granularity interaction proposed in GCA has effectively improved the localization effect by enhancing single-modal representation. At the same time, increasing the multi-granularity interaction of queries can enhance the model's ability to perceive fine-grained boundaries, i.e., improve the $R1@0.7$ by **6.10%**.

- **Effectiveness of different loss.** The final loss contains three parts: global loss, frame-moment correspondence loss and moment-frame correspondence loss. To evaluate the effectiveness of them, we remove some of them for comparison and the results are shown in Table 3, and we find that scale correspondence learning can inspire the potential of the model for boundary perception. In the presence of global loss, the frame-moment correspondence loss and moment-frame correspondence loss alone can enhance performance. The combination of the two can produce comprehensive performance improvement, and the $R1@0.7$ even gets more than 10% improvement than only using global loss, showing that scale correspondence learning can enhance the ability to capture key information and boundary perception.

### 4.4.2 Parameter sensitivity.

- **The mean weighted factor $\alpha$ in GCA.** An important hyperparameter in GCA is $\alpha$, which is used when combining the different granularities. It represents how much information is interacted and $\alpha = 1$ represents merely using global feature information while $\alpha = 0$ denotes only utilizing the fine granularity information. The specific results are shown in Figure 3(c). It can be observed that each metric shows a trend of first rising and then falling. Moreover, different difficulty of tasks represent diverse $\alpha$: for coarse-grained localization task ($m = 0.3$), $\alpha = 0.5$ is suitable while for fine-grained localization task, i.e., $m = 0.5$ and $m = 0.7$, $\alpha = 0.3$ is better.

- **The importance of frame-moment correspondence loss and moment-frame correspondence loss.** In the Eqn. (17), $\beta$ and $\gamma$ represent the importance of frame-moment correspondence loss and moment-frame correspondence loss in the final loss. We test different combinations of $\beta$ and $\gamma$ in the range

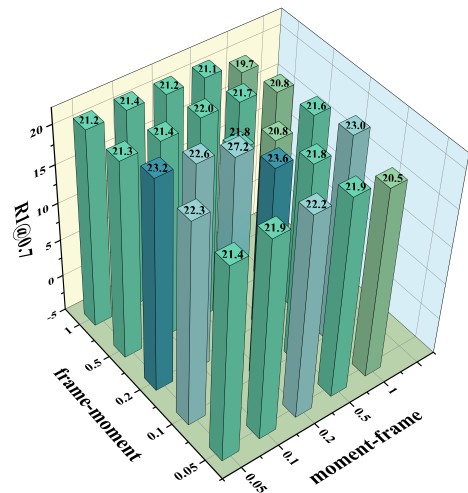

**Figure 4: The results of R1@0.7 from different frame-moment correspondence loss and moment-frame correspondence loss. Evaluation performed on the Charades-STA dataset.**

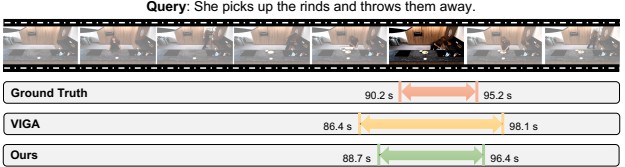

**Figure 5: Visualization of localization results on TACoS. The yellow bar represents the ground truth temporal boundaries of the language query, the blue bar depicts the predicted boundaries of ViGA, and the green bar signifies the predicted boundaries of SG-SCI.**

{0.05, 0.1, 0.2, 0.5, 1}, and display the numerical relationship between $R1@0.7$, as shown in Figure 4. Similar to the $\alpha$ in GCA, $\beta$ and $\gamma$ also show an overall trend of first increasing and then decreasing, and the highest value is obtained at $\beta = \gamma = 0.1$. The experimental results indicate that frame-moment correspondence loss and moment-frame correspondence loss have a significant impact on fine-grained boundary perception.

- **Sliding Window Size.** Sliding window is designed to capture multi-granularity features via multi-scale sliding windows. In fact, monotonically increasing or decreasing the size of the sliding window would lead to degradation of the representation discrimination, thus affecting overall performance. For instance, in extreme cases, the sliding window may encompass information from either the entire video or just a frame. In such scenarios the model struggles to learn and comprehend these moments holistically, leading to a decline in overall localization performance. We conducted ablation experiments on the Charades-STA dataset, with a default step size of 4. The specific experimental results are shown in Table 4. The results indicate that the best performance is achieved when the sliding window size is 8, indicating that the model needs a moderate sliding window size for sufficient learning during Scale Correspondence Learning.

**Table 4: Performance comparison on Charades-STA with different sliding window sizes, and $^*$ means the stride is the half of sliding window size.**

| Size | R1@0.3 | R1@0.5 | R1@0.7 | mIoU |
|------|--------|--------|--------|------|
| 4 | 70.89 | 49.46 | 23.66 | 46.02 |
| 8 | 70.30 | 52.07 | 27.23 | 46.77 |
| 16 | 70.78 | 47.18 | 21.99 | 45.23 |
| 24 | 70.43 | 46.02 | 21.48 | 44.96 |
| 32 | 69.84 | 45.3 | 19.46 | 44.22 |
| 4* | 70.86 | 49.46 | 23.31 | 45.91 |
| 16* | 70.83 | 47.15 | 22.07 | 45.22 |
| 24* | 70.46 | 45.86 | 21.64 | 44.70 |
| 32* | 69.09 | 44.44 | 20.16 | 44.03 |

**Table 5: Speed comparison on Charades-STA and TACoS between ViGA [10] and SG-SCI, time is averaged.**

| Method | Charades-STA | | TACoS | |
|--------|------|-----------|------|-----------|
| | Train | Inference | Train | Inference |
| ViGA [10] | 1.0x | 1.0x | 1.0x | 1.0x |
| SG-SCI | 1.1x | 1.2x | 1.0x | 1.2x |

*4.4.3 Inference speed.* As Table 5 illustrated, during the training and inference phases, our runtime is almost identical to ViGA's [10]. This is because our proposed GCA and SCL do not introduce significant computational complexity. As a result, we consider our methods to be competitive in terms of efficiency.

*4.4.4 Quantitative Results.* To more thoroughly examine the contributions of our proposed SG-SCI framework, we present an example illustrating the results of moment retrieval on the TACoS datasets. As is shown from the Figure 5, SG-SCI demonstrates superior effectiveness in terms of boundary perception. This can be attributed to the training process, where our model effectively enhances the model's perceptual ability across different scales.

# 5 CONCLUSION AND FUTURE WORK

In this paper, we designed a new framework to understand the correspondence connections between video and text. This framework used a limited amount of data from single frame to learn. Initially, we looked at the problems existing methods had in linking the meaning of video and text that varied in detail. To address these issues, our method included two main strategies: (1) creating a model that used underlying knowledge to connect features of different modalities, and (2) developing a learning strategy that focused on the differences in information between similar samples. These strategies helped our method to clearly define and find the differences in meaning across various details and modalities. This made our method better at representing features of explicit granularity and implicit scales. Experiments conducted on two benchmark datasets demonstrated the effectiveness of our proposed framework.

In the future, in order to further improve the performance of VML, we plan to introduce a more advanced visual-language transformer backbone. In addition, inspired by the fully-supervised paradigm, we plan to study the impact of different inference methods on localization performance.

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
