# OpenReview forum: "Explicit Granularity and Implicit Scale Correspondence Learning for Point-Supervised Video Moment Localization"
_acmmm.org/ACMMM/2024/Conference — MM2024 Poster_

### Official Review · Reviewer_GH41 · 2024-05-23

**Rating:** 4
**Confidence:** 3

**Summary:**

This paper introduces framework to understand the correspondence connections between video and text. It models semantic relations of different feature granularities and adaptively mines the implicit semantic scale to enhance modal feature representations.

**Strengths:**

The paper is well organized and easy to follow. The motivation is clear.

**Limitations:**

1. The proposed method contains several modules and involves several hyper-parameters, increasing the difficulty of the learning process.

2. The time consumption is just a relative value and I worry about whether this method can be applied in practice.

3. Why the Scale Correspondence Learning works?

4. I notice that, in Tab. 1 and Tab. 2, SG-SCI achieves good performance except the R1@0.3 in Tab. 1. It doesn't seem to make sense.

**Suitability:**

3

---

### Official Review · Reviewer_K1ei · 2024-05-27

**Rating:** 4
**Confidence:** 3

**Summary:**

The paper introduces the SG-SCI framework for point-supervised Video Moment Localization (VML), addressing challenges in explicit granularity alignment and implicit scale perception. It uses a Granularity Correspondence Alignment module and a Scale Correspondence Learning strategy to improve the alignment between video frames and textual descriptions, enhancing model performance for VML tasks. Experiments show the framework's effectiveness over existing methods.

**Strengths:**

1. The paper is well-written and well-motivated.
2. The solution to implicit scale perception and explicit granularity alignment is neat and novel.
3. The ablation study is thorough and the performance gain is significant shown in Table 3.

**Limitations:**

1. I recommend to simplify Figure 2. Current version is too implicit and not easy to follow the core idea of this paper.
2. Is there any study to test the robustness of the proposed method? Since the L159 claims that the robustness is enhanced.

**Suitability:**

2

---

### Official Review · Reviewer_igbp · 2024-06-03

**Rating:** 3
**Confidence:** 3

**Summary:**

The paper introduces the SG-SCI framework, which significantly improves point-supervised VML.

Key contributions:

	1.	Granularity Correspondence Alignment (GCA): Enhances semantic alignment between video frames and text descriptions.
	2.	Scale Correspondence Learning (SCL): Adapts to varying temporal scales, improving query matching accuracy.
	3.	Framework Design: Combines feature extraction, multi-modal interaction, and moment localization using sliding windows and Gaussian distribution.

Validated on Charades-STA and TACoS datasets, SG-SCI outperforms existing methods, showcasing its effectiveness in point-supervised video moment localization. This paper makes a notable contribution by addressing granularity alignment and scale perception challenges.

**Strengths:**

1. The paper does not use very advanced feature extractors such as CLIP, big foundation models (LLaVA, etc) but still achieves  SOTA results.
2. Achieve SOTA results and surpass the previous work by a large margin on a complex dataset.
3. The pipeline is somewhat novel.

**Limitations:**

1.	The overall approach lacks novelty. The proposed granularity correspondence alignment essentially employs attention mechanisms and cosine similarity, which are well-established techniques.
2.	The paper presents results primarily on two datasets. However, on the Charades-STA dataset, under the point-supervised setting, it fails to achieve state-of-the-art (SOTA) results for R1@0.3. Moreover, the second-best result cited is from 2022, indicating that the current SOTA result for this dataset dates back two years.
3.	Despite conducting extensive analytical experiments, the paper does not perform sufficient experiments on a diverse range of datasets to robustly demonstrate the generalizability and effectiveness of its main findings.

**Suitability:**

3

---

### Official Review · Reviewer_Kch1 · 2024-06-03

**Rating:** 4
**Confidence:** 2

**Summary:**

This paper proposes a granular correspondence alignment module to improve the correspondence relation across varying granularities and proposes a scale correspondence learning strategy. Extensive experiments are constructed to verify the effectiveness of the proposed methods.

**Strengths:**

1. The writing is easy to follow.
2. Extensive experiments demonstrate the effectiveness of the proposed methods.
3. The proposed methods are interesting.

**Limitations:**

1. The framework shown in Figure 2 seems to be complex.
2. The author should construct more ablation studies to compare the proposed methods (correspondence alignment module and the scale correspondence learning strategy) with other related methods.
3. It is suggested to construct the computational cost experiments.

**Suitability:**

3

---

### Official Review · Reviewer_i5qw · 2024-06-06

**Rating:** 5
**Confidence:** 2

**Summary:**

This work proposes an explicit granularity and implicit scale correspondence learning framework for point-supervised video moment localization. It aims to model the semantic alignment between video and text.

**Strengths:**

1. The basic idea, explicitly modeling semantic relations of different feature granularities and adaptively mining the implicit semantic scale is good.
2. The experimental results look good.

**Limitations:**

I cannot see the significant weakness of this work. I will just provide some minor suggestions:
1. Figure 2 should clearly demonstrate the interaction between model and strategy learning, to support the claim in the introduction.
2. While Figure 5 qualitatively shows the advantage of the proposed method, it would be better to show some failure cases and analyze the limitations.
3. Some relevant works should be discussed, e.g.
Luo et al., Soc: Semantic-assisted object cluster for referring video object segmentation, NIPS,2023

**Suitability:**

3

---

### Meta-Review · Area_Chair_va6Q · 2024-07-01

**Recommendation:** Accept (Poster)
**Confidence:** 5

**Metareview:**

This paper receives one weak accept, three borderline accepts, and one borderline reject initially. The reviewers raise some questions regarding the approach novelty and the experiment results (Sub-item indicator is not SOTA). Most of these questions are addressed in the rebuttal and recognized by reviewers. Eventually with the merit of the work, this paper receives one weak accept, three borderline accepts, and one borderline accept (initial score without final update), with agreement for acceptance. Considering that most sub-item indicators are SOTA with only one exception, AC recommended to accept. Authors are encouraged to revise the paper according to the reviews, and add an explanation/analysis for the underperformed sub-item indicator.